# Shift Aggregate Extract Networks

**Francesco Orsini**[12]**, Daniele Baracchi**[2] **and Paolo Frasconi**[2]

[1]Department of Computer Science
Katholieke Universiteit Leuven
Celestijnenlaan 200A
3001 Heverlee, Belgium
`francesco.orsini@kuleuven.be`

[2]Department of Information Engineering
Università degli Studi di Firenze
Via di Santa Marta 3
I-50139 Firenze, Italy
`daniele.baracchi@unifi.it`
`paolo.frasconi@unifi.it`

## Abstract

The Shift Aggregate Extract Network (SAEN) is an architecture for learning representations on social network data. SAEN decomposes input graphs into hierarchies made of multiple strata of objects. Vector representations of each object are learnt by applying *shift*, *aggregate* and *extract* operations on the vector representations of its parts. We propose an algorithm for domain compression which takes advantage of symmetries in hierarchical decompositions to reduce the memory usage and obtain significant speedups. Our method is empirically evaluated on real world social network datasets, outperforming the current state of the art.

## 1 Introduction

Many different problems in various fields of science require the classification of *structured data*, i.e. collections of objects bond together by some kind of relation. A natural way to represent such structures is through graphs, which are able to encode both the individual objects composing the collection (as vertices) and the relationships between them (as edges). A number of approaches to the graph classification problem has been studied in graph kernel and neural network literature.

Graph kernels decompose input graphs in substructures such as shortest paths (Borgwardt & Kriegel, 2005), graphlets (Shervashidze et al., 2009) or neighborhood subgraph pairs (Costa & De Grave, 2010). The similarity between two graphs is then computed by comparing the respective sets of parts. Methods based on recursive neural networks unfold a neural network over input graphs and learn vector representations of their nodes employing backpropagation though structure (Goller & Kuchler, 1996). Recursive neural networks have been successfully applied to domains such as natural language (Socher et al., 2011) and biology (Vullo & Frasconi, 2004; Baldi & Pollastri, 2003). An advantage of recursive neural networks over graph kernels, is that the vector representations of the input graphs are learnt rather than handcrafted.

Learning on social network data can be considerably hard due to their peculiar structure: as opposed to chemical compounds and parse trees, the structure of social network graphs is highly irregular. Indeed in social networks it is common to have nodes in the same graph whose degree differs by orders of magnitude. This poses a significant challenge for the substructure matching approach used by some graph kernels as the variability in connectivity generates a large number of unique patterns leading to diagonally dominant kernel matrices.

We propose Shift Aggregate Extract Networks (SAEN), a neural network architecture for learning representations of input graphs. SAEN decomposes input graphs into $\mathcal{H}$-hierarchies made of multiple strata of objects. Objects in each stratum are connected by "part-of" relations to the objects to the stratum above.

In case we wish to classify graphs we can use an $\mathcal{H}$-hierarchical decomposition in which the top stratum contains the graph $G$ that we want to classify, while the intermediate strata contain subgraphs of $G$, subgraphs of subgraphs of $G$ and so on, until we reach the bottom stratum which contains the vertices $v$ of $G$.

Unlike $\mathcal{R}$-convolution relations in kernel methods (which decompose objects into the set of their parts), $\mathcal{H}$-hierarchical decompositions are deep as they can represent the parts of the parts of an object.

Recursive neural networks associate to the vertices of the input graphs vector representations imposing that they have identical dimensions. Moreover, the propagation follows the edge connectivity and weights are shared over the whole input graph. If we consider that vector representations of nodes (whose number of parents can differ by orders of magnitude) must share the same weights, learning on social network data with recursive neural networks might be nontrivial.

SAEN compensates the limitations of recursive neural networks by adding the following degrees of flexibility:

1. the SAEN computation schema unfolds a neural network over $\mathcal{H}$-decompositions instead of the input graph,
2. SAEN imposes weight sharing and fixed size of the learnt vector representations on a per stratum basis instead of globally.

Indeed SAEN allows to use vector representations of different sizes for different strata of objects (e.g. graphs, subgraphs, subgraphs of subgraphs, edges, vertices etc.) The SAEN schema computes the vector representation of each object by applying *shift*, *aggregate* and *extract* operations on the vector representations of its parts.

Another contribution of this paper is the introduction of a domain compression algorithm, that we use in our experiments to reduce memory usage and runtime. Domain compression collapses objects in the same stratum of an $\mathcal{H}$-hierarchical decomposition into a compressed one whenever these objects are indistinguishable for the SAEN computation schema. In particular objects made of the same sets of parts are indistinguishable. In order obtain a lossless compression an $\mathcal{H}$-hierarchical decomposition we store counts on symmetries adopting some mathematical results from lifted linear programming (Mladenov et al., 2012). The domain compression algorithm is also reminiscent of the work of Sperduti & Starita (1997) in which common substructures of recursive neural networks are collapsed in order to reduce the computational cost.

## 2 SHIFT-AGGREGATE-EXTRACT NEURAL NETWORKS

We propose a neural network architecture that takes as input an undirected attributed graph $G = (V, E, X)$ where $V$ is the vertex set, $E \subseteq V \times V$ is the edge set, and $X = \{\mathbf{x}_v \in \mathbb{R}^p\}_{v \in V}$ is a set of $p$-dimensional vertex attributes. When vertices do not have associated attributes (for example this happens in some of the social network datasets of § 4.1), we can set $\mathbf{x}_v$ to some vertex invariant such as node centrality or betweenness.

### 2.1 $\mathcal{H}$-HIERARCHICAL DECOMPOSITIONS

Most graph kernels decompose graphs into parts by using an $\mathcal{R}$-convolution relation (Haussler, 1999). We extend this approach by decomposing graphs into a *hierarchy* of $\pi$-parametrized "part of" relations. Formally, an $\mathcal{H}$-hierarchical decomposition is a pair $(\{S_l\}_{l=0}^L, \{\mathcal{R}_{l,\pi}\}_{l=1}^L)$ where:

• $\{S_l\}_{l=0}^L$ are disjoint sets of objects $S_l$ called strata, or levels of the hierarchy. The bottom stratum $S_0$ contains non-decomposable objects (e.g. individual vertices), while the other strata $S_l$, $l = 1, \ldots, L$ contain composite objects, $o_i \in S_l$, whose parts $o_j \in S_{l-1}$ belong to the preceding stratum, $S_{l-1}$.

• $\{\mathcal{R}_{l,\pi}\}_{l=1}^L$ is a set of $l, \pi$-parametrized $\mathcal{R}_{l,\pi}$-convolution relations. A pair $(o_i, o_j) \in S_l \times S_{l-1}$ belongs to $\mathcal{R}_{l,\pi}$ iff "$o_j$ is part of $o_i$ with membership type $\pi$". For notational convenience, the parts of $o_i$ are denoted as $\mathcal{R}_{l,\pi}^{-1}(o_i) = \{o_j | (o_j, o_i) \in \mathcal{R}_{l,\pi}\}$.

The membership type $\pi$ is used to represent the roles of the parts of an object. For example, we could decompose a graph as a multiset of $\pi$-neighborhood subgraphs [1] in which $\pi$ is the radius of the neighborhoods (see Figure 1 on the left). Another possible use of the $\pi$ membership type is to

---

[1]The $r$-neighborhood subgraph (or ego graph) of a vertex $v$ in a graph $G$ is the induced subgraph of $G$ consisting of all vertices whose shortest-path distance from $v$ is at most $r$.

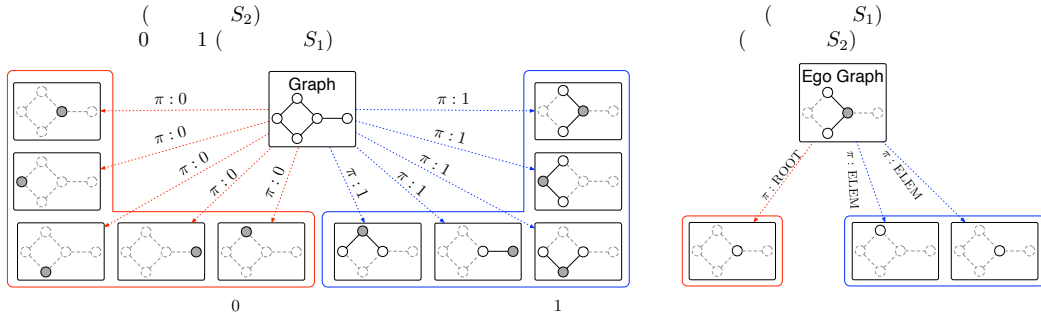

Figure 1: Image of an $\mathcal{H}$-hierarchical decomposition (in particular the EGNN explained in § 4.2). On the left we decompose a graph into rooted ego graphs of radius 0 and 1, while on the right we decompose an ego graph into the set of its vertices. The directed arrows represent "part of" relations labeled with their membership type $\pi$. The membership type $\pi$ represents the radius $\pi = 0, 1$ of the ego graphs (decomposition on the left) and the role (i.e. $\pi = $ ROOT, ELEM) of a vertex in the ego graph (decomposition on the right) respectively.

distinguish the root from the other vertices in a rooted neighborhood subgraph (see Figure 1 on the right).

An $\mathcal{H}$-hierarchical decomposition is a multilevel generalization of $\mathcal{R}$-convolution relations, and it reduces to an $\mathcal{R}$-convolution relation for $L = 1$.

## 2.2 SHIFT AGGREGATE EXTRACT SCHEMA FOR LEARNING REPRESENTATIONS

We propose Shift Aggregate Extract Network (SAEN) to learn vector representations for all the objects of all the strata $\{S_l\}_{l=0}^{L}$ in an $\mathcal{H}$-hierarchical decomposition. SAEN unfolds a neural network architecture over an $\mathcal{H}$-hierarchical decomposition by using the Shift Aggregate Extract (SAE) schema.

According to the SAE schema the vector representation of each object in the $\mathcal{H}$-hierarchical decomposition is either computed by applying a neural network on the vertex attributes (for the objects in bottom stratum) or defined in terms of the vector representations of its parts (for the other objects).

More formally, the SAE schema associates a $d_l$-dimensional representation $\mathbf{h}_i \in \mathbb{R}^{d_l}$ to each object $o_i \in S_l$ of the $\mathcal{H}$-hierarchical decomposition according to the following formula:

$$
\mathbf{h}_i = \begin{cases} f_0(\mathbf{x}_{v_i}; \Theta_0) & \text{if } o_i \in S_0 \\ f_l\underbrace{\left( \underbrace{\sum_{\pi \in \Pi_l} \sum_{o_j \in \mathcal{R}_{l,\pi}^{-1}(o_i)} \underbrace{(\mathbf{z}_\pi \otimes \mathbf{h}_j)}_{Shift}}_{Aggregate}; \Theta_l \right)}_{Extract} & \text{otherwise} \end{cases} \tag{1}
$$

where $f_l(\cdot; \Theta_l)$, $l = 0, \dots, L$ are multilayer neural networks with parameters $\Theta_l$.

With respect to the base case (first branch of Eq. 1) we have that each object $o_i$ in the bottom stratum $S_0$ is in one-to-one correspondence with the vertices $v_i \in V$ of the graph that we are decomposing. Indeed the vector representations $\mathbf{h}_i$ are computed by evaluating $f_0(\cdot; \Theta_0)$ in correspondence of the vertex attributes $\mathbf{x}_{v_i} \in X$.

The recursion step (second branch of Eq. 1) follows the Shift Aggregate Extract (SAE) schema:

- **Shift**: each part representation $\mathbf{h}_j \in \mathbb{R}^{d}_{l-1}$ is remapped into a space $\mathbb{R}^{|\Pi_l d_{l-1}|}$ made of $|\Pi_l|$ slots, where each slot has dimension $d_{l-1}$. This transformation shifts part representations $\mathbf{h}_j$ by using the Kronecker product $\otimes$ between an indicator vector $\mathbf{z}_\pi \in \mathbb{R}^{|\Pi_l|}$ and the vector representation $\mathbf{h}_j$ of part $o_j \in S_{l-1}$. The indicator vector $\mathbf{z}_\pi \in \mathbb{R}^{|\Pi_l|}$ defined as $z_i = \begin{cases} 1 & \text{if } i=\pi \\ 0 & \text{otherwise} \end{cases}$ and it is used to

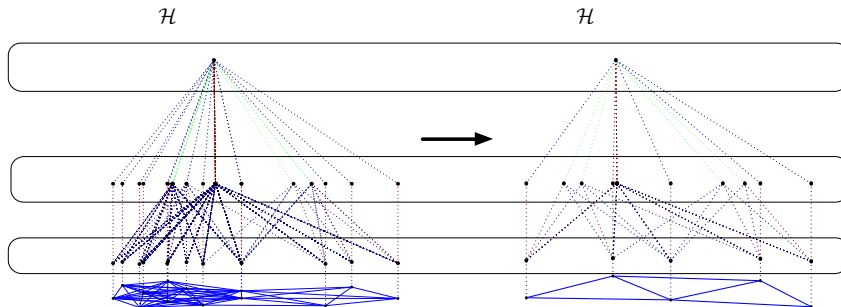

Figure 2: Pictorial representation of the $\mathcal{H}$-hierarchical decomposition of a graph taken from the IMDB-BINARY dataset (see § 4.1) together with its compressed version.

make sure that vector representations $\mathbf{h}_j$ of object parts will fall in the same slot if and only if they have the same membership type $\pi$.

- **Aggregate**: the shifted representations $(\mathbf{z}_\pi \otimes \mathbf{h}_j)$ of the parts $o_j$ are then aggregated with a sum.
- **Extract**: the aggregated representation is compressed to a $d_l$-dimensional space by a $\Theta_l$-parametrized nonlinear map $f_l(\cdot, \Theta_l) : \mathbb{R}^{|\Pi_l d_{l-1}|} \to \mathbb{R}^{d_l}$ implemented with a multilayer neural network.

The shift and aggregate steps, that we have seen so far, are identical to those used in kernel design when computing the explicit feature of a kernel $k(x, z)$ derived from a sum $\sum_{\pi \in \Pi} k_\pi(x, z)$ of base kernels $k_\pi(x, z)$, $\pi \in \Pi$. In principle, it would be indeed possible to turn SAEN into a kernel method by removing the extraction step E from the SAE schema. However, such an approach would increase the dimensionality of the feature space by a multiplicative factor $|\Pi_l|$ for each level $l$ of the $\mathcal{H}$-hierarchical decomposition, thus leading to an exponential number of features. When using SAEN, the feature space growth is prevented by exploiting a distributed representation (via a multilayered neural network) during the E step of the SAE schema. As a result, SAEN can easily cope with $\mathcal{H}$-hierarchical decompositions consisting of multiple strata.

## 2.3 EXPLOITING SYMMETRIES FOR DOMAIN COMPRESSION

In this section we propose a technique, called *domain compression*, which allows to save memory and speedup the SAEN computation. Domain compression exploits symmetries in $\mathcal{H}$-hierarchical decompositions by collapsing equivalent objects in each stratum. The greater the number of collapsed objects the highest the compression ratio.

Two objects $a$, $b$ in a stratum $S_l$ are collapsable $a \sim b$ if they share the same representation (i.e. $\mathbf{h}_a = \mathbf{h}_b$) for all the possible values of $\Theta_l$. A compressed stratum $S_l^{comp}$ is the quotient set $S_l/\sim$ of stratum $S_l$ w.r.t. the collapsibility relation $\sim$. We assume that the attributes of the elements in the bottom stratum $S_0$ are categorical, so that the same vector representation can be shared by multiple elements with non-zero probability. [2] While objects in the bottom stratum $S_0$ are collapsable when their attributes are identical, for all the other strata $S_l$, $l = 1, \ldots, L$, objects are collapsable if they are made by the same sets of parts for all the membership types $\pi$.

In Figure 2 we provide a pictorial representation of the domain compression of an $\mathcal{H}$-hierarchical decomposition (EGNN, described in § 4.2). On the left we show the $\mathcal{H}$-hierarchical decomposition of a graph taken from the IMDB-BINARY dataset (see § 4.1) together with its compressed version on the right.

### 2.3.1 DOMAIN COMPRESSION ALGORITHM

In order to compress $\mathcal{H}$-hierarchical decompositions we adapt the lifted linear programming technique proposed by Mladenov et al. (2012) to the SAEN architecture. If a matrix $M \in \mathbb{R}^{n \times p}$ has

---

[2] Vectors of real valued attributes could be discretized using clustering techniques. However, we leave discretization in SAEN to future works.

$m \leq n$ distinct rows it can be decomposed as the product $DM^{comp}$ where $M^{comp}$ is a compressed version of $M$ in which the distinct rows of $M$ appear exactly once. The Boolean decompression matrix, $D$, encodes the collapsibility relation among the rows of $M$ so that $D_{ij} = 1$ iff the $i^{th}$ row of $M$ falls in the equivalence class $j$ of $\sim$. A pseudo-inverse $C$ of $D$ can be computed by dividing the rows of $D^\top$ by their sum (where $D^\top$ is the transpose of $D$).

**Example 1** *If we look at matrix $M$ in Eq. 2 we notice that row $1$ and $4$ share the encoding $[0,0,0]$, rows $3$ and $5$ share the encoding $[1,1,0]$ while the encoding $[1,0,1]$ appears only once at row $2$. Matrix $M^{comp}$ is the compressed version of $M$.*

$$M = \begin{bmatrix} 0 & 0 & 0 \\ 1 & 0 & 1 \\ 1 & 1 & 0 \\ 0 & 0 & 0 \\ 1 & 1 & 0 \end{bmatrix} \quad M^{comp} = \begin{bmatrix} 0 & 0 & 0 \\ 1 & 0 & 1 \\ 1 & 1 & 0 \end{bmatrix} \quad D = \begin{bmatrix} 1 & 0 & 0 \\ 0 & 1 & 0 \\ 0 & 0 & 1 \\ 1 & 0 & 0 \\ 0 & 0 & 1 \end{bmatrix} \quad C = \begin{bmatrix} ^1/_2 & 0 & 0 & ^1/_2 & 0 \\ 0 & 1 & 0 & 0 & 0 \\ 0 & 0 & ^1/_2 & 0 & ^1/_2 \end{bmatrix} \quad (2)$$

*Matrix $M$ can be expressed as the matrix product between the decompression matrix $D$ and the compressed version of $M^{comp}$ (i.e. $M = DM^{comp}$), while the matrix multiplication between the compression matrix $C$ and the $M$ leads to the compressed matrix $M^{comp}$ (i.e. $M^{comp} = CM$).*

To apply domain compression we rewrite Eq. 1 in matrix form as follows:

$$H_l = \begin{cases} \underbrace{f_0(X; \Theta_0)}_{|S_0| \times d_0} & \text{if } l = 0 \\ \underbrace{f_l \left( \underbrace{\left[ R_{l,1}, \ldots, R_{l,\pi}, \ldots, R_{l,|\Pi_l|} \right]}_{|S_l| \times |\Pi_l||S_{l-1}|} \underbrace{\begin{bmatrix} H_{l-1} & \ldots & 0 \\ \vdots & \ddots & \vdots \\ 0 & \ldots & H_{l-1} \end{bmatrix}}_{|\Pi_l||S_{l-1}| \times |\Pi_l|d_{l-1}}; \Theta_l \right)}_{|S_l| \times d_l} & \text{otherwise} \end{cases} \quad (3)$$

where:

• $H_l \in \mathbb{R}^{|S_l| \times d_l}$ is the matrix that represents the $d_l$-dimensional encodings of the objects in $S_l$. The rows of $H_l$ are the vector representations $\mathbf{h}_i$ in Eq. 1, while the rows of $H_{l-1}$ are the vector representations $\mathbf{h}_j$ in Eq. 1;

• $X \in \mathbb{R}^{|S_0| \times p}$ is the matrix that represents the $p$-dimensional encodings of the vertex attributes in $V$ (i.e. the rows of $X$ are the $\mathbf{x}_{v_i}$ of Eq. 1);

• $f_l(\cdot; \Theta_l)$ is unchanged w.r.t. Eq. 1 and is applied to its input matrices row-wise;

• $R_{l,\pi} \in \mathbb{R}^{|S_l| \times |S_{l-1}|} \, \forall \pi \in \Pi_l$ are the matrix representations of the $\mathcal{R}_{l,\pi}$-convolution relations of Eq. 1 whose elements are $(R_{l,\pi})_{ij} = 1$ if $(o_j, o_i) \in \mathcal{R}_{l,\pi}$ and 0 otherwise.

Domain compression on Eq. 3 is performed by the DOMAIN-COMPRESSION procedure (see Algorithm 3) that takes as input the attribute matrix $X$ and the part-of matrices $R_{l,\pi}$ and returns their compressed versions $X^{comp}$ and the $R_{l,\pi}^{comp}$ respectively. The algorithm starts by invoking (line 1) the procedure COMPUTE-CD on $X$ to obtain the compression and decompression matrices $C_0$ and $D_0$ respectively. The compression matrix $C_0$ is used to compress $X$ (line 2) then we start iterating over the levels $l = 0, \ldots, L$ of the $\mathcal{H}$-hierarchical decomposition (line 4) and compress the $R_{l,\pi}$ matrices. The compression of the $R_{l,\pi}$ matrices is done by right-multiplying them by the decompression matrix $D_{l-1}$ of the previous level $l-1$ (line 5). In this way we collapse the parts of relation $\mathcal{R}_{l,\pi}$ (i.e. the columns of $R_{l,\pi}$) as these were identified in stratum $S_{l-1}$ as identical objects (i.e. those objects corresponding to the rows of $X$ or $R_{l-1,\pi}$ collapsed during the previous step). The result is a list $R^{col\text{-}comp} = [R_{l,\pi}D_{l-1}, \forall \pi = 1, \ldots, |\Pi_l|]$ of column compressed $R_{l,\pi}$−matrices. We proceed collapsing equivalent objects in stratum $S_l$, i.e. those made of identical sets of parts: we find symmetries in $R^{col\text{-}comp}$ by invoking COMPUTE-CD (line 6) and obtain a new pair $C_l$, $D_l$ of compression, and decompression matrices respectively. Finally the compression matrix $C_l$ is applied to the column-compressed matrices in $R^{col\text{-}comp}$ in order to obtain the $\Pi_l$ compressed matrices

DOMAIN-COMPRESSION$(X, R)$

```
1   C_0, D_0 = COMPUTE-CD(X)
2   X^comp = C_0 X  // Compress the X matrix.
3   R^comp = {}  // Initialize an empty container for compressed matrices.
4   for l = 1 to L
5       R^col_comp = [R_{l,π} D_{l-1}, ∀π = 1, …, |Π_l|]  // column compression
6       C_l, D_l = COMPUTE-CD(R^col_comp)
7       for π = 1 to |Π_l|
8           R^comp_{l,π} = C_l R^col_comp_π  // row compression
9   return X^comp, R^comp
```

Figure 3: DOMAIN-COMPRESSION

of stratum $S_l$ (line 8). Algorithm 3 allows us to compute the domain compressed version of Eq. 3 which can be obtained by replacing: $X$ with $X^{comp} = C_0 X$, $R_{l,\pi}$ with $R^{comp}_{l,\pi} = C_l R_{l,\pi} D_{l-1}$ and $H_l$ with $H^{comp}_l$. Willing to recover the original encodings $H_l$ we just need to employ the decompression matrix $D_l$ on the compressed encodings $H^{comp}_l$, indeed $H_l = D_l H^{comp}_l$.

As we can see by substituting $S_l$ with $S^{comp}_l$, the more are the symmetries (i.e. when $|S^{comp}_l| \ll |S_l|$) the greater the domain compression will be.

# 3 RELATED WORKS

When learning with graph inputs two fundamental design aspects that must be taken into account are: the choice of the pattern generator and the choice of the matching operator. The former decomposes the graph input in substructures while the latter allows to compare the substructures.

Among the patterns considered from the graph kernel literature we have paths, shortest paths, walks (Kashima et al., 2003), subtrees (Ramon & Gärtner, 2003; Shervashidze et al., 2011) and neighborhood subgraphs (Costa & De Grave, 2010). The similarity between graphs $G$ and $G'$ is computed by counting the number of matches between their common the substructures (i.e. a kernel on the sets of the substructures). The match between two substructures can be defined by using graph isomorphism or some other weaker graph invariant.

When the number of substructures to enumerate is infinite or exponential with the size of the graph (perhaps this is the case for random walks and shortest paths respectively) the kernel between the two graphs is computed without generating an explicit feature map. Learning with an implicit feature map is not scalable as it has a space complexity quadratic in the number of training examples (because we need to store in memory the gram matrix).

Other graph kernels such as the Weisfeiler-Lehman Subtree Kernel (WLST) (Shervashidze et al., 2011) and the Neighborhood Subgraph Pairwise Distance Kernel (NSPDK) (Costa & De Grave, 2010) deliberately choose a pattern generator that scales polynomially and produces an explicit feature map. However the vector representations produced by WLST and NSPDK are handcrafted and not learned.

A recent work by Yanardag & Vishwanathan (2015) proposes to uses pattern generators such as graphlets, shortest paths and WLST subtrees to transform input graphs into documents. The generated substructures are then treated as words and embedded in the Euclidean space with a CBOW or a Skip-gram model. The deep upgrade of existing graph kernels is performed by reweighing the counts of the substructures by the square root of their word-vector self similarity.

Another recent work by Niepert et al. (2016) upgrades the convolutional neural networks CNNs for images to graphs. While the receptive field of a CNN is usually a square window (Niepert et al., 2016) employ neighborhood subgraphs as receptive fields. As nodes in graphs do not have a specific temporal or spatial order, (Niepert et al., 2016) employ vertex invariants to impose an order on the nodes of the subgraphs/receptive fields.

## 4 EXPERIMENTAL EVALUATION

We answer to the following experimental questions:
**Q1** How does SAEN compare to the state of the art?
**Q2** Can SAEN exploit symmetries in social networks to reduce the memory usage and the runtime?

### 4.1 DATASETS

In order to answer the experimental questions we tested our method on six publicly available datasets first proposed by Yanardag & Vishwanathan (2015).

- **COLLAB** is a dataset where each graph represent the ego-network of a researcher, and the task is to determine the field of study of the researcher between *High Energy Physics*, *Condensed Matter Physics* and *Astro Physics*.
- **IMDB-BINARY**, **IMDB-MULTI** are datasets derived from IMDB where in each graph the vertices represent actors/actresses and the edges connect people which have performed in the same movie. Collaboration graphs are generated from movies belonging to genres *Action* and *Romance* for IMDB-BINARYand *Comedy*, *Romance* and *Sci-Fi* for IMDB-MULTI, and for each actor/actress in those genres an ego-graph is extracted. The task is to identify the genre from which the ego-graph has been generated.
- **REDDIT-BINARY**, **REDDIT-MULTI5K**, **REDDIT-MULTI12K** are datasets where each graph is derived from a discussion thread from Reddit. In those datasets each vertex represent a distinct user and two users are connected by an edge if one of them has responded to a post of the other in that discussion. The task in REDDIT-BINARYis to discriminate between threads originating from a discussion-based subreddit (*TrollXChromosomes*, *atheism*) or from a question/answers-based subreddit (*IAmA*, *AskReddit*). The task in REDDIT-MULTI5Kand REDDIT-MULTI12Kis a multi-class classification problem where each graph is labeled with the subreddit where it has originated (*worldnews, videos, AdviceAnimals, aww, mildlyinteresting* for REDDIT-MULTI5Kand *AskReddit, AdviceAnimals, atheism, aww, IAmA, mildlyinteresting, Showerthoughts, videos, todayilearned, worldnews, TrollXChromosomes* for REDDIT-MULTI12K).

### 4.2 EXPERIMENTS

In our experiments we chose an $\mathcal{H}$-hierarchical decomposition called Ego Graph Neural Network (EGNN), that mimics the graph kernel NSPDK with the distance parameter set to $0$.

Before applying EGNN we turn unattributed graphs $(V, E)$ into attributed graphs $(V, E, X)$ by annotating their vertices $v \in V$ with attributes $\mathbf{x}_v \in X$. We label vertices $v$ of $G$ with their degree and encode this information into the attributes $\mathbf{x}_v$ by employing the 1-hot encoding.

EGNN decomposes attributed graphs $G = (V, E, X)$ into a 3 level $\mathcal{H}$-hierarchical decomposition with the following strata (see Figure 1 for a pictorial representation of EGNN):
- stratum $S_0$ contains objects $o_v$ that are in one-to-one correspondence with the vertices $v \in V$.
- stratum $S_1$ contains $v_{root}$-rooted $r$-neighborhood subgraphs (i.e. ego graphs) $e = (v_{root}, V_e, E_e)$ of radius $r = 0, 1, \ldots, R$ and has part-of alphabet $\Pi_1 = \{\text{ROOT}, \text{ELEM}\}$. Objects $o_v \in S_0$ are "ELEM-part-of" ego graph $e$ if $v \in V_e \setminus \{v_{root}\}$, while the are "ROOT-part-of" ego graph $e$ if $v = v_{root}$.
- stratum $S_2$ contains the graph $G$ that we want to classify and has part-of alphabet $\Pi_2 = \{0, 1\}$ which correspond to the radius of the ego graphs $e \in S_1$ of which $G$ is made of.

**E1** We experimented with SAEN applying the EGNN $\mathcal{H}$-decomposition on all the datasets. For each dataset, we manually chose the parameters of SAEN, i.e. the number of hidden layers for each stratum, the size of each layer and the maximum radius $R$. We used the Leaky ReLU (Maas et al.) activation function on all the units. We report the chosen parameters in Table A1 of the appendix. In all our experiments we trained the neural networks by using the Adam algorithm to minimize a cross entropy loss.

The classification accuracy of SAEN was measured with 10-times 10-fold cross-validation. We manually chose the number of layers and units for each level of the part-of decomposition; the number of epochs was chosen manually for each dataset and we kept the same value for all the 100 runs of the 10-times 10-fold cross-validation.

Figure 4: Comparison of accuracy results.

| DATASET | DGK (Yanardag et al. 2015) | PSCN (Niepert et al., 2016) | SAEN (our method) |
|---|---|---|---|
| COLLAB | $73.09 \pm 0.25$ | $72.60 \pm 2.16$ | $\mathbf{75.63 \pm 0.31}$ |
| IMDB-BINARY | $66.96 \pm 0.56$ | $\mathbf{71.00 \pm 2.29}$ | $\mathbf{71.26 \pm 0.74}$ |
| IMDB-MULTI | $44.55 \pm 0.52$ | $45.23 \pm 2.84$ | $\mathbf{49.11 \pm 0.64}$ |
| REDDIT-BINARY | $78.04 \pm 0.39$ | $\mathbf{86.30 \pm 1.58}$ | $\mathbf{86.08 \pm 0.53}$ |
| REDDIT-MULTI5K | $41.27 \pm 0.18$ | $49.10 \pm 0.70$ | $\mathbf{52.24 \pm 0.38}$ |
| REDDIT-MULTI12K | $32.22 \pm 0.10$ | $41.32 \pm 0.42$ | $\mathbf{46.72 \pm 0.23}$ |

Figure 5: Comparison of accuracy on bio-informatics datasets.

| DATASET | PSCN ($k = 10^E$) (Niepert et al., 2016) | SAEN (our method) |
|---|---|---|
| MUTAG | $92.63 \pm 4.21$ | $84.99 \pm 1.82$ |
| PTC | $60.00 \pm 4.82$ | $57.04 \pm 1.30$ |
| NCI1 | $78.59 \pm 1.89$ | $77.80 \pm 0.42$ |
| PROTEINS | $75.89 \pm 2.76$ | $75.31 \pm 0.70$ |
| D&D | $77.12 \pm 2.41$ | $77.69 \pm 0.96$ |

The mean accuracies and their standard deviations obtained by our method are reported in Table 4, where we compare these results with those obtained by Yanardag & Vishwanathan (2015) and by Niepert et al. (2016).

Although our method was conceived for social network data, it can also handle other types of graphs. For the sake of completeness in Table 5 we report the mean accuracies obtained with SAEN on the molecule and protein datasets studied in previous works (e.g. Niepert et al. (2016)).

Table 1: Comparison of sizes and runtimes of the datasets before and after the compression.

| DATASET | SIZE (MB) | | | RUNTIME | | |
|---|---|---|---|---|---|---|
| | ORIGINAL | COMP. | RATIO | ORIGINAL | COMP. | SPEEDUP |
| COLLAB | 1190 | 448 | 0.38 | 43' 18" | 8' 20" | 5.2 |
| IMDB-BINARY | 68 | 34 | 0.50 | 3' 9" | 0' 30" | 6.3 |
| IMDB-MULTI | 74 | 40 | 0.54 | 7' 41" | 1' 54" | 4.0 |
| REDDIT-BINARY | 326 | 56 | 0.17 | TO | 2' 35" | $\geq 100.0$ |
| REDDIT-MULTI5K | 952 | 162 | 0.17 | OOM | 9' 51" | – |
| REDDIT-MULTI12K | 1788 | 347 | 0.19 | OOM | 29' 55" | – |

**E2** In Table 1 we show the file sizes of the preprocessed datasets before and after the compression together with the data compression ratio. [3] We also estimate the benefit of the relational compression from a computational time point of view and report the measurement of the runtime for 1 run with and without compression together with the speedup factor.

For the purpose of this experiment, all tests were run on a computer with two 8-cores Intel Xeon E5-2665 processors and 94 GB RAM. Uncompressed datasets which exhausted our server's memory during the test are marked as "OOM" (out of memory) in the table, while those who exceeded the time limit of 100 times the time needed for the uncompressed version are marked as "TO" (timeout).

## 4.3 DISCUSSION

**A1** As shown in Table 4, EGNN performs consistently better than the other two methods on all the social network datasets. This confirms that the chosen $\mathcal{H}$-hierarchical decomposition is effective on this kind of problems. Also the results for molecule and protein datasets (see Table 5) are in line with the current state of the art.
**A2** The compression algorithm has proven to be effective in improving the computational cost of our method. Most of the datasets improved their runtimes by a factor of at least 4 while maintaining the

---

[3]The size of the uncompressed files are shown for the sole purpose of computing the data compression ratio. Indeed the last version of our code compresses the files on the fly.

same expressive power. Moreover, experiments on REDDIT-MULTI5K and REDDIT-MULTI12K have only been possible thanks to the size reduction operated by the algorithm as the script exhausted the memory while executing the training step on the uncompressed files.

## 5 CONCLUSIONS

We proposed SAEN, a novel architecture for learning vector representations of $\mathcal{H}$-decompositions of input graphs. We applied SAEN for graph classification on 6 real world social network datasets, outperforming the current state of the art on 4 of them and obtaining state-of-the-art classification accuracy on the others. Another important contribution of this paper is the domain compression algorithm which greatly reduces memory usage and allowed us to speedup the training time of a factor of at least 4.

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

# APPENDIX: SHIFT AGGREGATE EXTRACT NETWORKS

**Francesco Orsini**[12]**, Daniele Baracchi**[2] **and Paolo Frasconi**[2]

[1]Department of Computer Science
Katholieke Universiteit Leuven
Celestijnenlaan 200A
3001 Heverlee, Belgium
`francesco.orsini@kuleuven.be`

[2]Department of Information Engineering
Università degli Studi di Firenze
Via di Santa Marta 3
I-50139 Firenze, Italy
`daniele.baracchi@unifi.it`
`paolo.frasconi@unifi.it`

## A PARAMETERS USED IN THE EXPERIMENTS WITH EGNN

In Table A1 we report for each dataset: the radiuses $r$ of the neighborhood subgraphs used in the EGNN decomposition and the number of units in the hidden layers for each stratum.

Figure A1: Parameters for the neural networks used in the experiments.

| DATASET | RADIUSES | HIDDEN UNITS | | |
|---|---|---|---|---|
| | $r$ | $S_0$ | $S_1$ | $S_2$ |
| COLLAB | $0, 1$ | $15 - 5$ | $5 - 2$ | $5 - 3$ |
| IMDB-BINARY | $0, 1, 2$ | $2$ | $5 - 2$ | $5 - 3 - 1$ |
| IMDB-MULTI | $0, 1, 2$ | $2$ | $5 - 2$ | $5 - 3$ |
| REDDIT-BINARY | $0, 1$ | $10 - 5$ | $5 - 2$ | $5 - 3 - 1$ |
| REDDIT-MULTI5K | $0, 1$ | $10$ | $10$ | $6 - 5$ |
| REDDIT-MULTI12K | $0, 1$ | $10$ | $10$ | $20 - 11$ |
| MUTAG | $0, 1, 2, 3$ | $10$ | $5 - 5$ | $5 - 5 - 1$ |
| PTC | $0, 1$ | $15$ | $15$ | $15 - 1$ |
| NCI1 | $0, 1, 2, 3$ | $15$ | $15$ | $15 - 10 - 1$ |
| PROTEINS | $0, 1, 2, 3$ | $3 - 2$ | $6 - 5 - 4$ | $6 - 3 - 1$ |
| D&D | $0, 1, 2, 3$ | $10$ | $5 - 2$ | $5 - 3 - 1$ |

