# Peer review of "Shift Aggregate Extract Networks"

_ICLR 2017 — rejected_

[Official Review · AnonReviewer3 · rating 3 · confidence 2 · 16 Dec 2016]
**Might be something good here, but key details are missing.**

Some of the key details in this paper are very poorly explained or not even explained at all. The model sounds interesting and there may be something good here, but it should not be published in it's current form. 

Specific comments:

The description of the R_l,pi convolutions in Section 2.1 was unclear. Specifically, I wasn't confident that I understood what the labels pi represented.

The description of the SAEN structure in section 2.2 was worded poorly. My understanding, based on Equation 1, is that the 'shift' operation is simply a summation of the representations of the member objects, and that the 'aggregate' operation simply concatenates the representations from multiple relations.  In the 'shift' step, it seems more appropriate to average over the object's member's representations h_j, rather than sum over them.

The compression technique presented in Section 2.3 requires that multiple objects at a level have the same representation. Why would this ever occur, given that the representations are real valued and high-dimensional? The text is unintelligible: "two objects are equivalent if they are made by same sets of parts for all the pi-parameterizations of the R_l,pi decomposition relation." 

The 'ego graph patterns' in Figure 1 and 'Ego Graph  Neural Network' used in the experiments are never explained in the text, and no references are given. Because of this, I cannot comment on the quality of the experiments.

[Official Review · AnonReviewer2 · rating 5 · confidence 3 · 19 Dec 2016]
**Interesting approach, confusing presentation.**

The paper contributes to recent work investigating how neural networks can be used on graph-structured data. As far as I can tell, the proposed approach is the following:

  1. Construct a hierarchical set of "objects" within the graph. Each object consists of multiple "parts" from the set of objects in the level below. There are potentially different ways a part can be part of an object (the different \pi labels), which I would maybe call "membership types". In the experiments, the objects at the bottom level are vertices. At the next level they are radius 0 (just a vertex?) and radius 1 neighborhoods around each vertex, and the membership types here are either "root", or "element" (depending on whether a vertex is the center of the neighborhood or a neighbor). At the top level there is one object consisting of all of these neighborhoods, with membership types of "radius 0 neighborhood" (isn't this still just a vertex?) or "radius 1 neighborhood".

  2. Every object has a representation. Each vertex's representation is a one-hot encoding of its degree. To construct an object's representation at the next level, the following scheme is employed:

    a. For each object, sum the representation of all of its parts having the same membership type.
    b. Concatenate the sums obtained from different membership types.
    c. Pass this vector through a multi-layer neural net.

I've provided this summary mainly because the description in the paper itself is somewhat hard to follow, and relevant details are scattered throughout the text, so I'd like to verify that my understanding is correct.

Some additional questions I have that weren't clear from the text: how many layers and hidden units were used? What are the dimensionalities of the representations used at each layer? How is final classification performed? What is the motivation for the chosen "ego-graph" representation? 

The proposed approach is interesting and novel, the compression technique appears effective, and the results seem compelling. However, the clarity and structure of the writing is quite poor. It took me a while to figure out what was going on---the initial description is provided without any illustrative examples, and it required jumping around the paper to figure for example how the \pi labels are actually used. Important details around network architecture aren't provided, and very little in the way of motivation is given for many of the choices made. Were other choices of decomposition/object-part structures investigated, given the generality of the shift-aggregate-extract formulation? What motivated the choice of "ego-graphs"? Why one-hot degrees for the initial attributes?

Overall, I think the paper contains a useful contribution on a technical level, but the presentation needs to be significantly cleaned up before I can recommend acceptance.

[Author Response · Francesco Orsini · 28 Dec 2016]
**We provide a summary of the changes that we made after the reviewers' feedback.**

A) we rewrote section 2:
    A1) we generally improved the wording,
    A2) we added a figure in order to exemplify H-hierarchical decompositions,
    A3) we improved the explanation of the \pi labels (now called "membership types"),
    A4) we rewrote and clarified the assumptions under which domain compression works,
    A5) we added a footnote with the definition of r-neighborhood subgraphs/ego-graphs,
    A6) we introduced the term 'collapsibility' for the equivalence relation between objects in the same stratum.

B) We added (in section 4) the details about the network architecture and how the classification is performed.

C) We added an appendix with the number of layers and hidden units that we used in our experiments.

[Official Review · AnonReviewer1 · rating 5 · confidence 3 · 30 Dec 2016]
**Poor performance on bioinformatics dataset?**

the paper proposed a method mainly for graph classification. The proposal is to decompose graphs objects into hierarchies of small graphs followed by generating vector embeddings and aggregation using deep networks. 
The approach is reasonable and intuitive however, experiments do not show superiority of their approach. 

The proposed method outperforms Yanardag et al. 2015 and Niepert et al., 2016 on social networks graphs but are quite inferior to Niepert et al., 2016 on bio-informatics datasets. the authors did not report acccuracy for Yanardag et al. 2015 which on similar bio-ddatasets for example NCI1 is 80%, significantly better than achieved by the proposed method. The authors claim that their method is tailored for social networks graph more is not supported by good arguments? what models of graphs is this method more suitable?

[Final Decision · Program Chairs · 06 Feb 2017]
**ICLR committee final decision**

The authors present a novel architecture, called Shift Aggregate Extract Network (SAEN), for learning representations on social network data. SAEN decomposes input graphs into hierarchies made of multiple strata of objects. The proposed approach gives very promising experimental results on several real-world social network datasets. 
 
 The idea is novel and interesting. However, the exposition of the framework and the approach could be significantly improved. The authors clearly made an effort to revise the paper and improve the clarity. Yet, the paper would still certainly benefit from a major revision, to clarify the exposition and spell out all the details of the framework. 
 
 A extensive use of the space in the supplement could potentially help overcoming the space limitation in the main part of the paper which can make exposition of new frameworks challenging in general. This would allow an extensive explanation of the proposed framework and related concepts. 
 
 A major revision of the paper will generate a stronger submission, which we invite the authors to submit to the workshop track.